# Eco-Friendly Biocontrol Strategies of *Alternaria* Phytopathogen Fungus: A Focus on Gene-Editing Techniques

**Domingo Cesar Carrascal-Hernández** [1], **Edwin Flórez-López** [2], **Yeimmy Peralta-Ruiz** [3],
**Clemencia Chaves-López** [4] **and Carlos David Grande-Tovar** [1,*]

1 Grupo de Investigación en Fotoquímica y Fotobiología, Programa de Química, Facultad de Ciencias Básicas, Universidad del Atlántico, Puerto Colombia 081008, Colombia

2 Grupo de Investigación en Química y Biotecnología QUIBIO, Universidad Santiago de Cali, Calle 5 No 62-00, Cali 760035, Colombia

3 Facultad de Ingeniería, Programa de Ingeniería Agroindustrial, Universidad del Atlántico, Carrera 30 Número 8-49, Puerto Colombia 081008, Colombia

4 Faculty of Bioscience and Technology for Food, Agriculture and Environment, University of Teramo, Via R. Balzarini 1, 64100 Teramo, Italy

* Correspondence: carlosgrande@mail.uniatlantico.edu.co; Tel.: +57-5-3599484

**Abstract:** Agricultural food production is greatly affected by postharvest diseases worldwide, such as the diseases caused by *Alternaria* species, which are very common in several crops. The management of fungal infections around the world largely relies on fungicides. In this context, the control of diseases such as early blight caused by *Alternaria solani* in potatoes and *Alternaria linariae* in tomatoes has mainly consisted of the application of fungicides, with negative impacts on the environment and human health. Recently, the application of 'omics' and gene editing through the CRISPR/Cas9 system and RNAi technologies demonstrated their effectiveness as emerging greener alternatives for controlling phytopathogenic fungi. Additionally, coatings based on essential oils and microbial antagonists suggest alternative strategies for controlling phytopathogenic fungi that are respectful of the environment. This review presents an exhaustive literature review focused on using greener alternatives to the traditional management of postharvest diseases associated with *Alternaria* species, such as inhibiting pathogenicity from their phytopathogenic genes using gene editing based on CRISPR/Cas9 and RNAi technologies. The review also presents coatings based on essential oils and microbial antagonists as greener strategies for *Alternaria* control. Biological processes of maximum efficiency can replace chemical methods for controlling phytopathogenic fungi, preserving healthy conditions in agricultural lands and ecosystems. This is possible with the rise of 'omic' technologies, the CRISPR/Cas9 tool, and RNAi technology. Greener control methods of *Alternaria* fungi can increase agricultural production, improving the economy and global health.

**Keywords:** *Alternaria* control; CRISPR/Cas9; leaf spot; phytopathogenic genes; RNAi; postharvest disease

## 1. Introduction

The control of phytopathogenic fungi in agricultural production is of global interest since they cause significant economic losses to producers [1]. About six million fungi are known; however, about 200 species have some pathogenicity associated with diseases affecting biodiversity, ecology, agriculture, and food security worldwide [2]. Some estimates of losses caused by phytopathogenic fungi in crops are 21.5% for wheat, 30% for rice, 22.5% for corn, 17.2% for potatoes, and 21.4% for soybeans, especially in the postharvest production stage [3]. Within these fungi, *Alternaria* species are ubiquitous phytopathogenic fungi that cause deadly and common diseases in crops, including *Alternaria* leaf spot and *Alternaria* brown spot, characterized by necrotic spots on plants' vegetative tissue [4].

Additionally, these fungi are of concern since ingesting food contaminated by mycotoxins contributes to developing serious diseases such as cancer [5].

*Alternaria* species associated with common crop diseases have been characterized through conidial morphology, pathogenicity, and phylogenetic analysis [6]. In addition, it is possible to identify fungal species through the isolation, characterization, and identification of mycotoxins encoded by unique species of phytopathogenic fungi. It is also possible to strike toxin-encoding genes (TCGs) and chromosomes related to pathogenicity (also known as accessory chromosomes) through the use of 'omics' [7].

Many strategies have been proposed in order to control *Alternaria* diseases, such as chemicals fungicides, biocontrol methods, agro-nanotechnology, improving the resistance of plants to fungal diseases, and the obtention of new genotypes able to act as biocontrol agents (through the induction of genetic mutations in the virulent genotypes) providing new avirulent strains that can compete directly with the virulent ones [8]. In recent years, the study of TCGs such as polyketide synthase A (*PksA*), associated with protein biosynthesis, as well as the *AMT*, *AFT*, *AKT*, and *ALT* gene groups, associated with pathogenicity and host specificity in *A. alternata* [9], has allowed the development of strategies based on gene silencing or activation, taking advantage of the CRISPR/Cas9 system [10]. In this context, the silencing of genes associated with adaptive capacity has been reported in *Alternaria* species such as *Alternaria brassicicola*, as well as a significant reduction in the production of secondary metabolites (or mycotoxins), which exert a degradative effect on organelles essential for the cell, which generates necrotic spots in the vegetative tissue in an advanced stage of the disease [11]. Additionally, mutagenesis has been reported to inhibit both mycelial growth and mycotoxin production by *Alternaria alternata* strains, which has shown significant susceptibility in the fungal cell wall [12].

Biosynthetic genes are interesting in developing alternative pathogenicity control methods for *Alternaria* (and other phytopathogenic fungi), especially phytopathogenic genes, which are producers of various secondary metabolites (small molecules with biological activity in the organism) that are closely related to the various interactions with other organisms and are located (clustered) adjacently in the genomes of various plant pathogenic species [13]. These groups of genes associated with the production of secondary metabolites have been marked as targets for gene editing methodologies, given that these genes are related to the fungal chemodiversity of various phytopathogenic species, as well as the functional variety and horizontal transfer. These methodologies aim to replace chemical processes through genome editing based on CRISPR/Cas9, allowing the modification of genes with efficiencies greater than 50%, programmable base editing with varied programming between 29% and 63%, and activations of genes between 78% and 89% [14]. Additionally, methodological strategies based on RNA interference (RNAi) have proven to be valuable mechanisms of defense of plants against pathogens [15].

RNAi is a gene suppression mechanism conserved in eukaryotes, which consists mainly of three enzymes: RNA-dependent RNA polymerase (Rdrp), Dicers (Dcr), and Argonautes (Ago) [16]. De Oliveira Filho et al. (2021) discussed gene suppression through RNAi, which consists of the introduction of double-stranded RNA (dsRNA) molecules that include complementary target messenger RNA (mRNA) sequences into cells. Double-stranded RNA (dsRNA) is formed from single-stranded RNA (ssRNA) by the action of Rdrp. Subsequently, Dcr recognizes and cuts dsRNA into short fragments of sizes from 20–24 nucleotides, the ssRNA. The ssRNA then interacts with the complex protein components (RISC), which include Ago proteins that recognize the target mRNA producing RNA-induced silencing [17]. The cleavage of the target mRNA prevents the translation by activating the RISC that interacts directly with the genes of interest, forming a complex that eliminates RNA-mediated transcription [18].

Other eco-friendly alternatives for phytopathogenic fungi management include essential oils and microbial antagonists. Aslam et al. (2022) evaluated the effects of essential oils of *Thymus vulgaris* with a high efficiency against the mycelial growth of *A. alternata*, the agent of the rot in *Eriobotrya japonica* [19]. Essential oils are rich in secondary metabolites,

which have desirable biological activities and a high control potential in phytopathogenic fungi [20]. The use of microbial biocontrol agents in the control of postharvest diseases is also promising for the management of black spot disease caused by *A. alternata*, as reported in *Ziziphus jujuba*, through the antagonistic yeast *Debaryomyces nepalensis* [21].

This comprehensive review focuses on controlling the pathogenicity of causal species such as *Alternaria alternata*, *Alternaria tenuissima*, *Alternaria burnsii*, and *Alternaria brassicicola* through gene editing studies and using green alternatives such as essential oils and antagonist microbials. These alternatives have emerged with a potential for improving plant resistance, also contributing to controlling diseases caused by ingesting mycotoxins from contaminated food, and increasing the fertility of arable agricultural spaces, ultimately allowing the preservation of ecosystems.

## 2. *Alternaria* spp.

*Alternaria* is a genus of the phylum Ascomycota (family of Pleosporaceae, order of Pleosporales). It is one of the most ubiquitous fungi, with the characteristic of inhabiting any environment, given its adaptability to different substrates and minerals to grow [22]. *Alternaria* species are infectious fungi with a broad host diversity in crops. Resistance to multiple environmental conditions and genetic diversity give *Alternaria* species a great adaptation capacity in different areas of the world [23].

The description of the *Alternaria* genus dates to 1816, with identification of the first type of *Alternaria tenuis*. Since then, about a thousand *Alternaria* species have been reported [24].

Phylogenetic analyses have been employed in *Alternaria* through the study of ribosomal DNA, mtSSU, and protein-coding genes (glyceraldehyde-3-phosphate dehydrogenase (*gpd*), endopolygalacturonase, *Alternaria* allergen Alt a 1, beta-tubulin, translation elongation factor 1-alpha (*TEF* 1-α), RNA polymerase II (RPB2), calmodulin, chitin synthase, *Tsr*1, plasma membrane *ATPase*, actin, 1,3,8–trihydroxynaphthalene (*THN*)), using multilocus sequence analysis [25]. Recently, Ahmedpur et al. (2021) evaluated the diversity of *Alternaria* species in *Cyperaceae* and *Juncaceae* through multilocus sequence analysis of ITS, parts of genes belonging to allergen Alt a 1, *gpd*, RPB2, and the translation elongation *TEF* 1-α. The study revealed previously known species, such as *Alternaria scirpivola* and *Alternaria caricicola*, and three new species: *Alternaria cypericola* sp. nov., *Alternaria heyranica* sp. nov., as well as *Alternaria junciacuti* sp. nov. in plants with the symptoms of leaf spot [26]. Table 1 reports a sample of *Alternaria* species identified in crops causing leaf and black spot disease and their hosts.

**Table 1.** Examples of characterization studies in *Alternaria* species and the different hosts infected.

| Species | Hosts | Disease | Ref. |
|---|---|---|---|
| *A. alternata* (Fr) Keissl | Potato (*Solanum tuberosum*)<br>Tomato (*Lycopersicum esculentum*)<br>Pepper (*Capsicum annuum*)<br>Eggplant (*Solanum melongena*)<br>Pomegranate (*Punica granatum*) | Leaf spots | [27,28] |
| *A. brassicae* (Berk) Sacc. | Oilseeds<br>Vegetables<br>Canola (*Brassica napus*)<br>Mustard (*Brassica juncea*) | Black spots | [29] |
| *A. brassicicola* (Schwein) Wiltshire | Cabbage varieties<br>Mustard (*B. juncea*)<br>Potato (*Solanum tuberosum*)<br>Carrot (*Daucus carota* L.)<br>Tomato (*Lycopersicum esculentum*) | Black spots | [30] |

**Table 1.** *Cont.*

| Species | Hosts | Disease | Ref. |
|---|---|---|---|
| *A. dauci* (Kuhn) J. W. Groves y Akolko | Carrot (*Daucus carota* L.) Lettuce (*Lactuca sativa*) Celery (*Apium graveolens*) Spinach (*Spinacea oleracea*) | Round or oval white spots with brown edges 2–4mm in diameter (disease known as *Alternaria* leaf spot) | [31] |
| *A. infectoria* | Rice (*Oryza sativa*) Cereal varieties | Necrotic leaf lesions | [32] |
| *A. solani* (Ellis y G. Martin) L. R. Jones | Potato (*Solanum tuberosum*) | Necrotic leaf spots | [33] |
| *A. porri* (Ellis) Ciffer | Onion (*Allium cepa*) | Purple and black lesions | [34] |
| *A. tenuissima* (Nees y T. Nees: Fr) Wiltshire | Kiwifruit (*Actinidia* spp.) Wheat (*Triticum*) Apples (*Malus domestica*) Tomatoes (*Lycopersicum esculentum*) Blueberries (*Vaccinium myrtillus*) Pomegranate (*Punica granatum*) | Circular brown necrotic lesions up to 8 mm | [35–37] |

* Polymerase chain reaction (PCR).

## 2.1. Alternaria Infection

Diseases caused by *Alternaria* species, such as *Alternaria* leaf spot and *Alternaria* brown spot, are prevalent in agricultural production [38]. These diseases present necrotic spots on the leaves and vegetative tissues, as shown in Figure 1.

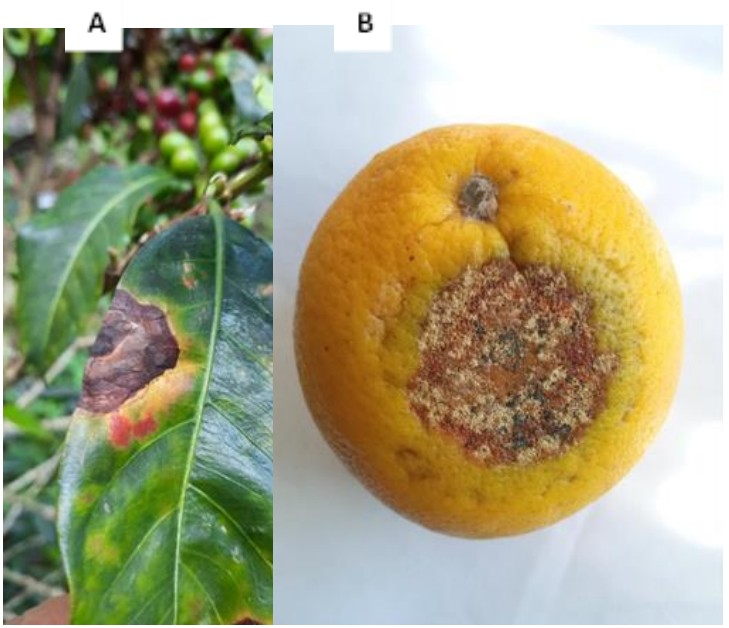

**Figure 1.** *Alternaria* leaf spot disease on a coffee plant (*Coffea Arabica*, **A**) and *Alternaria* brown spot disease on citrus (*Citrus sinensis*, **B**). This image shows the leaves' (**A**) and fruits' fungal lesions (**B**).

*Alternaria* species are endophytic, saprophytic, and pathogenic, related to diseases in preharvest and postharvest production [39]. The sporulation of these fungi facilitates propagation by the breeze and through vectors such as insects, which contract the spores of phytopathogenic fungi and spread them to healthy plants [40]. Advanced *Alternaria* infections in crops of economic importance generate putrefaction in postharvest production through cell wall degradation, metabolism of fatty acids, sugars, and the increase in secondary metabolites in their hosts, which are discarded in an advanced disease stage [41].

Infection with *Alternaria* species occurs through contact with the fungus spores. The penetration into the vegetative tissue usually occurs through plant injury or the routes of plant supplies, directly through the epidermis or via stomata. The *Alternaria* spores remain inactive on the vegetative tissue until they find some path to enter the host or recognize nutrients and minerals. Subsequently, a germination process starts through the secretion of enzymes and mycotoxins. In contrast, the host plants of these pathogens produce new substances (generally oligosaccharides) in response to the infection process, which slows down the spread of the infection. Oligosaccharides induce defense enzymes in plants, significantly improving resistance to diseases transmitted by the soil or pathogens [42].

The environments most prone to damage by *Alternaria* on plants are humid zones with a mild acid to neutral pH. The infection process usually begins in fruits at an advanced stage of maturity, where the defense system progressively weakens and tissues soften, while nutrient and pH conditions are ideal [43].

The pathogenicity and virulence depend on specific genes producing host-specific toxins (HSTs) and non-host-specific toxins (NHSTs). These genes are included in conditionally dispensable chromosomes (CDCs), presumably acquired by horizontal transfer from other pathogens [44,45]. Horizontal gene transfers in pathogenic species increase the range of existing pathogens [46]. In the case of *Alternaria* species, the pathogenicity is described by genes encoding both HSTs and NHSTs [47]. Meena et al. (2019) reported the molecular characterization of the toxins encoded by *Alternaria* species, including HSTs and NHSTs. Both HSTs and NHSTs are distributed in four families of toxins: the first belongs to epoxy-decatrienoic acid (EDA)-AK-AF and ACT; the second belongs to cyclic depsipeptide (cyclic tetrapeptide) AM toxin; the third family is the amino pento/polyketide (sphininganine analogs) AAL toxins; and finally, those belonging to polyketides—ACR toxins, which cause degeneration in the membranes of important cell organelles, such as chloroplasts, mitochondria, plasma membrane, Golgi complex, among others [48], as shown in Figure 2.

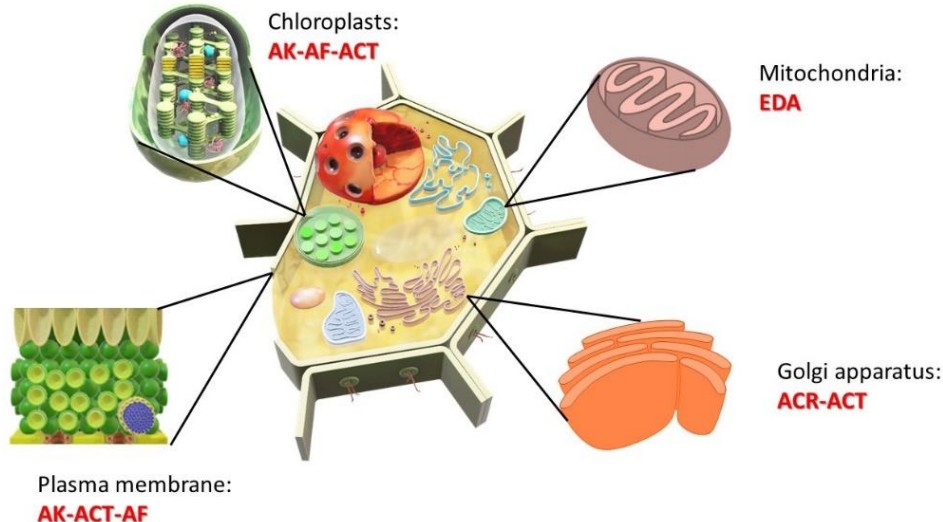

**Figure 2.** Primary targets (major membranous organelles) of host-specific toxins for cell damage.

The mechanisms associated with host-specific toxin production linked to the pathogenicity of *A. alternata* and *A. arborescens* have been reported by polymerase chain reaction (PCR) assays targeting three genes (AMT1, AMT2, and AMT14), which were identified in a conditionally expendable chromosome involved in the production of host-specific toxins [49].

The production of more than 70 toxins in *Alternaria* species is known and associated with pathogenicity. Several of them are harmful to human beings and animals [50]. Among them, alternariol (AOH), alternariol monomethyl ether (AME), tenuazonic acid (TA), altenuene (ALT), and tentoxin (TEN) are the most widely spread in foods and plants,

fully characterized by high-performance liquid chromatography coupled to mass spectrometry (HPLC-MS) [51]. The use of spectrometry (UPLC-MS/MS) to identify *Alternaria*-associated mycotoxins was reported in 260 medicinal and edible herbs showing symptoms of *Alternaria*-associated diseases. AME was the most prevalent mycotoxin with 37.5%, followed by AOH with 22.5%, ALT with 17.5%, TA with 10.83%, alterotoxin with 7.5%, and altenusin with 4.17% [52].

Although the identification of *Alternaria* species is carried out by molecular methods, the production of secondary metabolites has also been used for classification and identification. In this context, Saleem et al. (2022) reported the molecular characterization of *A. tenuissima*, *A. mimicula*, *A. infectoria*, *A. alternata,* and *A. arborescens* in tomatoes and the ability of these species to produce AOH AME, TA, and ALT, using HPLC. Additionally, they used PCR to amplify the ITS region and sequenced the *pksJ* and *pksH* genes associated with toxin production [53].

Previous research has suggested molecular variability between species, which presents different levels of virulence. Garganese et al. (2016) recorded a significant correlation between the pathogenicity of *Alternaria* and the expression of the *ACTT1, pksJ*, and *pksH* genes, associated with the synthesis of AOH, AME, TeA, and AlT. The expression of *ACTT1* depends specifically on the geographical zones of crops [54]. The study of mycotoxins secreted by *Alternaria* species is of interest since they allow the rapid identification of diseases in crops associated with single species of phytopathogenic fungi. Interspecies molecular variability presents a big problem for both agricultural producers and consumers. Still, it has expanded the field of study through the metabolic pathways, allowing the development of genetic assemblies and the prediction of genes through omics technologies [55].

## 2.2. Phytopathogenic Genes in Alternaria spp.

*Alternaria* species (like most crop pathogens) possess chromosomes known as expendable coding chromosomes (ECCs), which contain specific genes encoding HSTs. Gai et al. (2021) studied the HSTs produced by the ACT toxin-encoding gene cluster and reported the genome assembly of *A. alternata* strain Z7 at the chromosomal level. The genome has 34.24 Mb, 12067 protein-coding genes, 34 ribosomal RNAs (rRNAs), 107 transfer RNAs (tRNAs), and two ECCs, which offer information on the chromosomes directly linked to the pathogenicity of the organism. This is valuable because it allows an understanding of the pathogenicity mechanism of *A. alternata* [56].

Despite the little information available to study genes that encode toxins from *Alternaria* species, data mining offers possibilities to build an alternative to control pathogens through proteomic and transcriptomic analysis [57]. Using these technologies, whole genome sequencing was reported for *A. solani* (altNL03003) [58], the agent causing early blight disease in potato and tomato (genomic DNA sequencing ID: 4190556). Additionally, the genomes of other *Altermaroa* species have been sequenced and studied in depth [59].

The codominant *ASC-1* gene has been identified (also known as *ASC*, *ASC-1*, and *LASS* 2) using comparative genomics, which confers resistance to stem canker caused by the necrotrophic fungus *A. alternata f* sp. *lycopersicis* that affects *Solanum lycopersicum* [60]. The *ASC-1* gene is homologous to genes belonging to most eukaryotes (from yeast to humans) and has a conserved function. For example, a yeast strain with deletions in the *LAG1* and *LAC1* genes (homologous to the *ASC* gene) complemented the *ASC-1* gene [61]. It is known that the different toxins produced by *Alternaria* species vary in the site of action during infection. Still, all of them converge in the death cell of their hosts [62,63]. Figure 3 shows the homolog of the *ASC-1* gene (also reported in homologs of some strains of *Alternaria* and yeasts) located on chromosome three of *S. lycopersicum*, which can reveal details about the infection and resistance mechanisms by plants against attacks by phytopathogenic fungi [64].

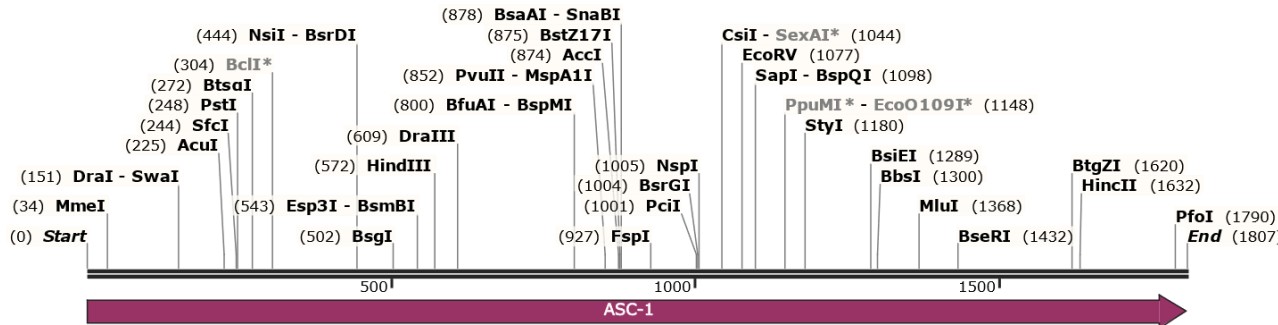

**Figure 3.** Co-predominant *ASC-1* gene identified in both *A. alternata* and *Septoria lycopersici*, pathogens that cause necrotic spots in plants, located on chromosome 3. * Methylation prohibited breakpoints similar to Dam methylation (*) [65].

The study of gene expression levels after 18 h of inoculation with *A. solani* in the breeding lines BD12-65 and BD1235 showed that the *ASC-1* gene is involved in the resistance response in the mentioned lines.

Studying the metabolic genes, also known as biosynthetic genes, associated with essential metabolic pathways for a species is also fascinating. These biosynthetic genes allow an understanding of the metabolite synthesis or the production of mycotoxins [66]. Figure 4 shows (in purple) the central biosynthetic genes of *A. alternata*, positioned in their genome (with sizes up to 6066 nt) and other genes (in gray) whose precise functions are unknown.

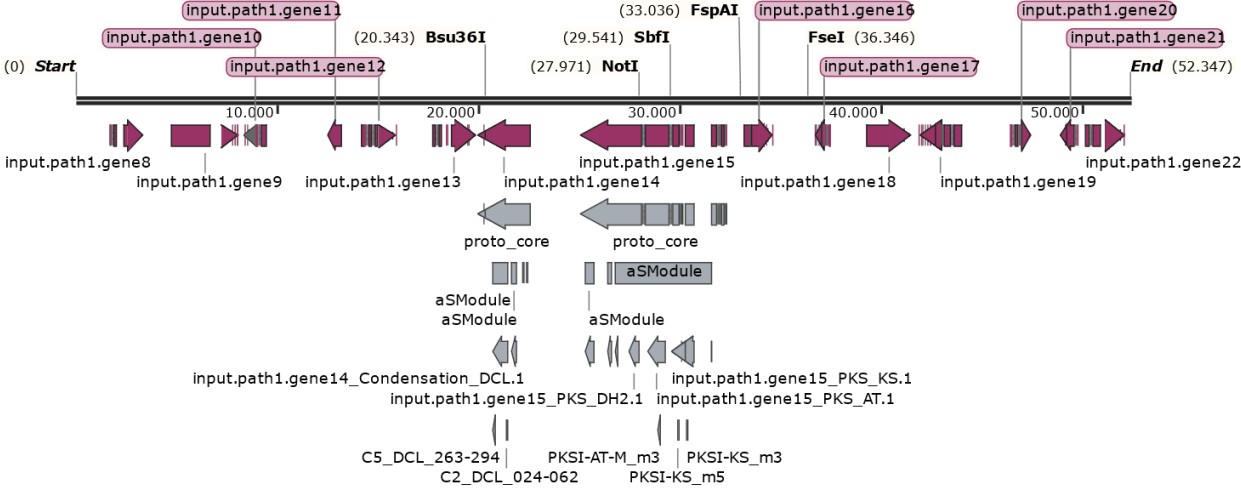

**Figure 4.** Biosynthetic genes of *A. alternata* and other genes whose function is unknown.

The elucidation of genes of interest in phytopathogenic fungi through in silico or in vitro studies, such as genes encoding mycotoxins, as in the case of *A. atra* (a saprophytic fungus producing leaf blight on the potato) and *A. arborescens* (producing stem blight) that affects tomato crops and is a producer of host-specific toxins, is of great interest to combat losses associated with diseases transmitted by phytopathogenic fungi since they offer the possibility of applying the gene-editing techniques such as CRISPR/Cas9 or RNAi [67,68].

In this context, Wenderoth et al. (2019) reported the genes encoding AOH and its derivative AME in *A. alternata* by CRISPR/Cas9-mediated gene knockout [69]. In addition, information on genes associated with synthesizing secondary metabolites was reported. The associated genes consist of a group of 15 kb, including a *pksI* (polyketide synthase gene), *omtI* (an O-methyltransferase), and *moxI* (a FAD-dependent monooxygenase), *sdrI* (short-chain dehydrogenase), *doxI* (extradiol dioxygenase response gene), *aohR* (transcription factor gene). The inhibition of the transcription factor gene *aohR* generated a significant

reduction in AOH levels, consequently decreasing the virulence in citrus, apple, and tomato crops [69].

Liu et al. (2020) reported the regulation of four genes, *PksJ, PksF, PksI,* and *PksA,* which are polyketide synthase type and associated with the synthesis of secondary metabolites [70]. The study suggests that the inhibition of *PksJ, PksF, PksI,* and *PksA* eliminates the biosynthesis of AOH, AME, and TEA in *A. alternata.*

### 3. *Alternaria* Control Strategies

#### 3.1. Chemical Control Strategies

The use of fungicides in agricultural fields is a widespread practice in the world. The excessive use of pesticides and the commercialization of food contaminated with fungicides poisons more than four million people yearly, with 350,000 lethally poisoned people. Additionally, the residues of fungicides in commercialized foods generate more than 20,000 cases of cancer per year [71]. Pereira et al. (2021) reported the countries where chemical compounds are used as the primary treatment to control pathogens, their incidence in the environment, and the adverse effects on non-target species [72].

Chemical control methods include compounds with dicarboximide groups such as promicidone (Prolex®, Promidona®, Proroc®, and Sideral®) [73,74], benzimidazole (Benlate®) [75], and triazole (epoxiconazole®, triadimenol®, propiconazole®, ciproconazole®, teboconazole®, and mefentrifluconazole®) [76]. They are associated with several diseases and increased cross-resistance. For example, cases of *A. alternata* held with mancozeb® and difenoconazole®, which are widely used in crops [77].

Additionally, the excessive use of these chemical techniques to control pathogens harms arable fields, causing a reduction in soil fertility.

#### 3.2. Green Control Strategies

##### 3.2.1. Essential Oils and Biopolymers Used to Control Alternaria Fungi

Biofungicides based on essential oils and biopolymers to control phytopathogenic fungi have been used as greener alternatives for managing crop pathogens [78]. Essential oils are of particular interest as an alternative for managing diseases transmitted by phytopathogenic fungi, given their efficiency, biodegradability, and non-toxicity in human health and the environment [79]. In recent years, several studies have demonstrated the efficacy of essential oils. For example, Feng et al. (2007) reported the inhibitory effects of essential oils from thyme (*Thymus vulgaris*), eucalyptus (*Eucalyptus globulus* Labill), cassia (*Cassia didymobotrya*), sage (*Salvia officinalis*), and nutmeg (*Myristica Fragans* Houtt) against *A. alternata*, the agent causing necrotic spots [80]. Zaker et al. (2010) demonstrated the antifungal capacity of eucalyptus (*Eucalyptus globulus*), peppermint (*Mentha piperita*), Lavandula (*Lavandula angustifolia*), and alcoholic extracts of datura against *A. alternata* [81]. Additionally, Huang et al. (2019) reported a solid antifungal capacity of *Artemisia lavandulaefolia* DC and *Artemisia scoparia* essential oils, while *Artemisia annua* oil presented a moderated activity against *A. solani* [82]. In the same way, the extract of *Syzygium aromaticus*, *Betle piper, Zingiber cassumunar,* and *Coscinium fenestratum* had inhibitory activity against *A. brassicicola* [83]. The controlled release of essential oils has consisted of the nanoencapsulation of essential oils with antifungal activity [84]. Antifungal lipid nanoemulsions incorporating essential oils have effectively controlled tomato leaf spots caused by *A. alternata*. In addition, lipid nanoemulsions encapsulating garlic (*Allium sativum*), chilli (*Capsicum annuum L. var annuum*), and cinnamon (*Cinnamomum zeylanicum*) essential oils exhibit high antifungal activity in vivo against *A. alternata* [85].

It is well known that the chemical structure of essential oil components is fundamental for disease control. For example, the essential oil of basil (*Ocimum basilicum*) is rich in compounds with antifungal activities (28.7% linalool and 38.2% methyl chavicol) [86], which has an inhibitory effect on conidia germination and growth of *A. alternata*. However, despite the chemical composition, several external factors limit the inhibitory efficiency of essential oils, such as humidity, light sensitivity, high temperatures, volatility, hydrophobicity, and

high oxygen content environments. These limitations are compensated by encapsulation or coatings with substances such as polymers that protect them from these unfavorable external effects [87]. The development of films or coatings used to preserve food consists of systems in which various biopolymers are combined to achieve beneficial properties. Usually, these coatings are made from proteins, lipids, and carbohydrates [88].

On the other hand, carbohydrate-based biopolymers such as alginate and chitosan are very attractive in protecting crops against fungi due to their antifungal properties and biodegradability [89]. Using chitosan (a derivative of deacetylated chitin found in the outer layer of crustaceans such as shrimp, crab, lobster, and shellfish) as a coating in crop protection against phytopathogenic fungi is well studied. It is considered an ideal resource for sustainable agriculture thanks to its biodegradability, non-toxicity, and biocompatibility properties [90]. Table 2 reports examples of free and encapsulated essential oils used to control *Alternaria* species in several hosts.

**Table 2.** Examples of free and encapsulated essential oils used to manage diseases caused by *Alternaria* species.

| Phytopathogenic Fungus | Disease Caused | Host/Type of Study | Source Of Essential Oil | Polymer Matrix * | Ref. |
|---|---|---|---|---|---|
| *A. alternata* | Leaf spot—*Alternaria* rot | *Berberis vulgaris; Lycopersicum esculentum;* in vivo and in vitro | *Carum carvi* L., *Thimus vulgaris* L., *Pimpinella anisum, Chamaemelum nobile, Origanum majorana* | - | [79] |
| | | In vivo and in vitro | *Trachyspermum copticum, Foeniculum vulgare* L., and *Carum carvi* | - | [91] |
| | | In vitro | *Echinophora platyloba* | - | [92] |
| | | - | *Origanum vulgare* L., *Thimus vulgaris* L. and *Syzygium aromaticum* | - | [93] |
| | | In vitro | *Thymus zygiis* | - | [94] |
| | | | | - | |
| | | Cherry tomato/in vitro and in vivo | *Laurus nobilis* L. | - | [95] |
| | | Cherry tomato/in vitro and in vivo | *Cymbopogon nardus* | - | [96] |
| | | | *Origanum onites* L., *Thymbra spicata* L., *Lavandula stoechas* L., subsp. *Stoechas* L., *F. vulgare* Mill, *Laurus nobilis* L. | - | [97] |
| | | | *Zanthoxylum armatum* DC | - | [98] |
| *A. solani* | Early blight | In vivo and in vitro | *Angelica archangelica* | - | [99] |
| *Alternaria* spp. | Alternariose— black leaf spot | *Malus domestica* in vivo and in vitro | *Pinus pinea* | | [100] |
| | | | *Pulicaria mauritanica* | | [101] |
| | | | *Warionia saharae* | | [102] |
| | | | *Cocos nucifera* and hydrogenated *Trachycarpus fortunei* oil | | [103] |
| *A. humicola* | Alternariose | In vivo and in vitro | *Asarum heterotropoides* | - | [104] |
| *A. tenuissima* | Early blight | In vivo and in vitro | *F. vulgari—Ocimum basilicum—Citrus lemmon—Rosmarinus officinalis—Salvia officinalis* | - | [105] |

**Table 2.** *Cont.*

| Phytopathogenic Fungus | Disease Caused | Host/Type of Study | Source Of Essential Oil | Polymer Matrix * | Ref. |
|---|---|---|---|---|---|
| *A. brassicicola* | Black leaf spot | In vivo and in vitro | *Polygonum perfoliatum* | - | [106] |
| *A. porri* | Early blight | In vivo and in vitro | *Cymbopocon citratus—Syzygium aromaticum—Cinnamomum aromaticum—Cinnamomum verum—Ocimum basilicum—Foeniculum vulgare, and Oenothera biennis* | - | [107] |
| *A. arborescens* | *Alternaria* rot | In vivo and in vitro | *O. vulgare—Thymus vulgaris—Cymbopogon citratus—Coriandrum sativum* | - | [108] |
| **Essential oils with Biopolymers** | | | | | |
| *A. alternata* | Leaf spot | In vitro | | CS **, CS-saponin NPs and CS-Cu NPs ** | [109] |
| | | In vivo and in vitro | *Byrsonima crassifolia* extract | CS NPs | [110] |
| *A. alternata* | *Alternaria* rot | Prunus avium | Nettle extract—spruce extract | CS—oligosaccharides | [111] |
| *Alternaria species* | *Alternaria* rot | In vivo and in vitro | - | CS combined with natamycin | [112] |
| *A. alternata* | *Alternaria* rot—leaf spot—early blight | In vivo and in vitro | Polyphenolic extracts from orange peel | - | [113] |
| *A. tenuissima* | Leaf spot | In vivo and in vitro | *Cicer arietinum* | Chickpea vicilin | [114] |

\* In case of encapsulation or coatings. ** CS—chitosan, NPs—nanoparticles.

### 3.2.2. Antagonistic Microorganisms to Control *Alternaria*

Research based on antagonist microorganisms (such as antagonistic rhizobacteria) in the control of phytopathogenic fungi is of interest to inhibit fungal growth and enhance the healthy development of plants. More environmentally friendly alternatives use biological fungal control methods by applying antagonistic bacterial and fungal genera of *Gliocladium*, *Bacillus*, *Coniothyrium*, *Paecilomices*, *Phlebiopsis*, *Pseudomonas*, and *Rhizobium*, among others [115], which are based on mycoparasitism or direct parasitism through bacteria. Bacterial-based parasitism has been studied in previous publications, which report that these associations between antagonistic microorganisms and phytopathogenic fungi are based on the adherence of antagonistic microorganisms to both the conidia and vegetative hyphae of the fungus, which aggressively compete for nutrients [116]. These bacteria have been identified as species belonging to the *Bacillus* genus, such as *B. cereus*, *B. mycide,* and *B. subtilis*, which have shown high efficiency in controlling *Alternaria helianthi*. In this context, Kurniawan et al. (2018) evaluated the control of mycelial growth of *A. alternata* through the use of antagonistic *Pseudomonas* and *Bacillus* bacteria, with a 27% inhibition of mycelial growth and 50% inhibition of postharvest rot in blueberry (*Vaccinium corymbosum*) development [117]. Additionally, antagonistic fungi to control phytopathogenic fungi are an alternative whose importance is increasing given the high percentages of inhibition of many pathogens. Komhorm et al. (2021) evaluated the antagonistic activity of *Talaromyces flavus*, *Talaromyces trachyspermus*, *Neosartorya fischeri*, *Eupenicillium* sp., and *Gongronella butleri* against black spot disease in *Brassica oleracea var. saberlica* caused by *A. brassicicola* [118]. Table 3 summarizes biological control agents such as fungi and bacteria.

The use of plant growth promoting rhizobacteria (PGPR), in addition to suppressing plant diseases, improves their growth, showing a 13% reduction in the severity of illness transmitted by *A. solani* and 84.3% protection in samples treated with *Achromobacter xylosoxidans*, *Lysinibacillus fusiformis*, and *Bacillus subtilis* [119]. Additionally, isolates of *Pseudomonas*

*aeruginosa* and *Enterobacter roggenkampii* exhibited inhibitory activity against *A. alternata.* In addition, postharvest treatment of tomato fruits using yeasts such as *Meyerozyma guilliermondii* showed a 57% reduction in lesions, while using *Pseudomonas aeruginosa* exhibited 50%, 60% with *E. roggenkampii,* and 50% with *M. guilliermondii* [120]. Sharma et al. (2018) studied the antagonistic activity of rhizobacterial isolates against *A. brassicae* blight in *Brassica juncea* L, revealing high disease inhibition [121].

**Table 3.** Examples of fungi and bacteria for the biological control of *Alternaria* species in crops.

| Pathogen | Disease | Host | Antagonist | Inhibition | Ref. |
|---|---|---|---|---|---|
| **Bacteria** | | | | | |
| *A. alternata* | Leaf spot | *Nicotiana tabacum* | *Bacillus megaterium* | 76% inhibition in mycelial growth after seven days | [122] |
| | | | *Pseudomonas corrugata* | 58% inhibition of mycelial growth after five days | [123] |
| *Alternaria Species* | Leaf spot | *Punica granatum* L. | *Bacillus subtilis—Bacillus pumilus—Bacillus amyloliquefaciens—Bacillus mojavensis—Bacillus vallismortis—Solibacillus silvestris—B. megaterium—Corynebacterium glutamicum—Erwinia herbicola—Pantoea dispersa—Bacillus cereus—Bacillus endophyticus* | *Bacillus mojavensis* presented the highest efficiency in mycelial growth inhibition with 80%. *Bacillus myloliquefaciens* 78.9%, *Bacillus vallismortis* 76.7%, and *B. subtilis* 75.6% | [124] |
| *A. brassicicola* | Leaf spot | *Brassica rapa* | *B. subtilis—Pseudomonas fluorescens—Streptomyces hydrogenans* | *B. subtilis* generated the highest inhibition of mycelial growth (24%) after seven days, followed by *Pseudomonas fluorescens* with 21.7% inhibition after 15 days and *Streptomyces hydrogenans* with 22.2% after 12 days | [125] |
| **Fungi** | | | | | |
| *A. tenuissima* | Early blight | *Solanum tuberosum* | *Wickerhamomyces anomalus* | *W. anomalus* inhibits infection by *A. tenuissima.* Combined with UV, it stimulates and improves biocontrol activity against the disease. Treatment with *W. anomalus*–UV generates a defense response in potatoes | [126] |
| *A. alternata* | Stem rot | *Actinidia cinensis* | *Fusicolla violacea (J-1)* | J-1 exhibited 66.1% inhibition under in vitro conditions | [127] |
| *A. alternata (CABI strain 353822), A. tenuissima (CABI strain 352931), and A. infectoria (CBS strain 120149)* | Leaf spot | *Oriza sativa* | *Pseudomonmas libanensis—Pseudomonas rhodesiae* | After inoculation of *Alternaria* species with a total of 110 *Pseudomonas* isolates, a slight reduction in fungal growth was recorded, with a significant decrease in mycotoxin production | [128] |

### 3.3. Molecular-Based Methods for Controlling Alternaria

Biosynthetic genes are interesting for developing alternative pathogenicity control methods for *Alternaria* and other phytopathogenic fungi. The alternative techniques intend to replace the use of chemical processes through genome editing strategies based on gene knockout, CRISPR/Cas9, and RNAi techniques [14].

Lu et al. (2019) reported the silencing of the *AbSte7* gene, involved in specific aspects of the pathogenicity of *A. brassicicola*, related to the improvement in the adaptation capacity and inhibition of the host's immune response [11]. The *AbSte7* gene's disruption allowed a considerable reduction in the enzymes involved in the host cell wall degradation during infection, suggesting a significant decrease in pathogenicity, opening possibilities for non-conventional biological methods' application in controlling phytopathogenic fungi for agro-industrial production.

### 3.3.1. Control of Alternaria Pathogenicity Based on CRISPR/Cas9

The activation of biosynthetic genes is of particular interest for controlling the pathogenicity of various fungi. Due to the complexity of the selective activation of genes of interest, few successful cases have been reported, which poses a challenge to studying the biosynthetic potential of genes of interest. However, some activation methodologies consisting of global regulators have been successful [129,130].

Methodological techniques based on CRISPR/Cas9 have significantly impacted biomedical research by correcting errors at the genome level through the activation or deactivation of specific genes. Applications of CRISPR/Cas9 have been reported in genome editing in filamentous fungi [131]. However, few studies are based on *Alternaria*, despite the well-studied *Aspergillus nidulans* [132,133]. CRISPR/Cas9 consists of two essential molecules: a guide RNA complementary to a gene of interest and an endonuclease that breaks specific bonds in the double helix chain, known as Cas proteins (or proteins associated with CRISPR) [134], as shown in Figure 5.

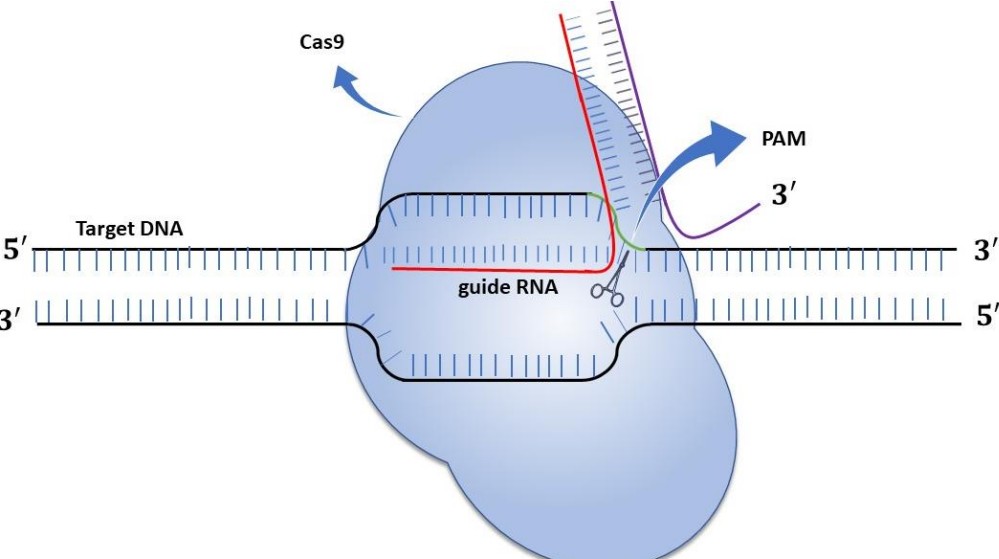

**Figure 5.** In the CRISPR/Cas9 system, the Cas9 endonucleases are directed to their target by a section of gRNA. The gRNA fragment that accompanies the DNA is between 18 and 20 nucleotides long. The cut consists of 2 and 6 nucleotides (the exact sequence depends on the Cas enzyme) located at the 3′ end of the gRNA, known as the protospacer adjacent motif (PAM).

The biosynthetic potential in phytopathogenic fungi is mainly unknown, which is why the number of reports of plasmids used in CRISPR/Cas9 is increasing, especially in studying fungal genes linked to the pathogenicity of filamentous fungi. In this regard, Roux et al. (2020) reported the activation of biosynthetic gene clusters in filamentous fungi by applying CRISPR/Cas9 to obtain the highest biosynthetic potential encoded in fungal genomes. The technique is based on a CRISPR/dLbCas12a-VPR system targeting the *micA* gene linked to non-ribosomal peptides similar to synthetases (similar to NRPS), allowing more significant production of microperfuranone and the discovery of dehydrocroperfuranone, demonstrating the utility of the dLbCas112aD156R-VPR variant for

CRISPR/Cas9 under culture conditions [135]. Previously, Nødvig et al. (2015) developed a methodology based on CRISPR/Cas9 for use in filamentous fungi based on gRNA-guided mutagenesis. The system consists of four CRISPR/Cas9 vectors equipped with fungal markers that facilitate selecting a wide range of fungi [132]. New CRISPR/Cas9 variants that are increasingly efficient in editing fungal genomes have been reported, such as the ribonucleoprotein-based CRISPR/Cas9 variant and methodologies based on the in vivo expression of Cas9 [136]. The establishment of genome editing mediated by the CRISPR/Cas9 system in economically important filamentous fungi is becoming increasingly relevant and exciting in industrial biotechnology and agriculture [137,138].

CRISPR-based gene editing systems such as the CRISPR interference system (CRISPRi) and CRISPR activation strategy (CRISPRa) are used for gene editing. These two systems, CRISPRi and CRISPRa, are used in gene editing strategies in phytopathogenic fungi [139]. Both methods and their variants have allowed a greater understanding of the genes of phytopathogenic species and their regulations through the manipulation of their genomes, also used in novel gene editing methodologies such as excision DNA, base editing, and epigenetic modifications [140,141].

CRISPR/Cas9 transcriptional activity is a powerful system for overexpression of target genes [142], whose efficiency is based on the enzymatic fusion or the transcriptional activation domain with dCas9; dCas9 is a variant of CRISPR/Cas9, consisting of an alteration in the domain of the Cas9 enzymes that decreases the efficiency in DNA breaks. dCas9-mediated transcriptional regulation occurs at the elongation or transcriptional points, reducing the amount of mRNA through competitive inhibition. Interference efficiencies of up to 90% have been reported in eukaryotic cells [143]. The complex formed with dCas9 facilitates the assembly of the RNA transcriptional machinery in a specific target guided by gRNA. This complex potentially increases the expression of genes associated with the biosynthesis of secondary metabolites [144].

The production of gene-associated host-specific metabolites is still largely unknown. However, developing new methodologies based on CRISPR/Cas9 may help elucidate the compounds produced by various genes and the subcellular localization of proteins in phytopathogenic fungi that regulate the transport of nutrients across multiple target fungi to inhibit their growth [145,146]. The CRISPR/Cas9 system has enabled targeted mutagenesis efficiently in plants to improve resistance to fungal diseases [147]. In this context, two different strategies have been used to induce disease resistance in plants with CRISPR/Cas9: (i) the pathogen–gene method, which encompasses engineering a sgRNA into the plant chromosome that directs Cas9 to target a specific pathogen gene, thus hampering pathogenesis, (ii) the plant–gene method which uses a sgRNA that targets endogenous plant genes involved in pathogen interactions and modifies them to either improve the host immune response, or to interfere with the host-recognition pathway of the pathogen [148].

CRISPR/Cas9-induced mutations of disease susceptibility (S) genes have generated resistance to pathogens, including fungi, oomycetes, bacteria, and viruses [149]. However, genome editing-mediated plant resistance to *Alternaria* species remains relatively unexploited. In a recent review, Dort et al. (2020) gave an overview of CRISPR/Cas systems and their adaptation to the CRISPR/Cas9 technology in plant pathology [147].

### 3.3.2. RNAi in the Control of *Alternaria*

Additionally, methodological strategies have been developed based on RNA interference (RNAi), which positively influenced the plants' defense mechanisms against various phytopathogenic fungi [15]. Genetic control methods are promising as emerging biocontrol alternatives through molecular studies, leading to genetic information of fungal species using 'omic' technologies [150]. RNAi is a biological gene silencing mechanism involving the degradation of specific sequences of mRNA mediated by double-stranded RNA, applied in functional gene research and genetic modification in plants. Gene suppression by target RNA silencing plays a critical role in studying phytopathogenic fungal control

and host-induced gene silencing through RNAi-generated signals in the plant [151]. Additionally, a spray gene silencing methodology has been reported that protects plants from diseases transmitted by phytopathogenic fungi through direct spraying of dsRNA-targeted resistance genes onto plant tissue, which is an excellent alternative to the use of synthetic or rudimentary fungicides [152].

The mechanism of host-induced gene silencing through RNAi-generated signals in the host plant requires dsRNA expression of specific sequences in the host plant and silencing of target genes [153]. The dsRNA expression implies the transformation of the host plant through the construction of a hairpin dsRNA containing the gene of the target pathogen. The degradation of the target pathogen mRNA occurs through the generation of small RNAi homologous to the target RNA, which protects plants from diseases transmitted by phytopathogenic fungi [154,155].

These induced resistance techniques are well studied and successfully applied to control various necrotrophic fungi and oomycetes through the expression of endogenous fungal genes to stimulate increased resistance to diseases transmitted by phytopathogenic fungi [156].

The techniques of resistance induced in host plants vary according to the variations in the RNAi pathways. RNAi pathways differ in the RNA sources and the mechanisms by which gene silencing is achieved. However, they all follow similar processing steps and are triggered by a dsRNA molecule. The processing of dsRNA generates RNA fragments of 21–25 bp (sRNA) through Dicer enzyme processing. Then, the fragment is incorporated into a protein complex known as the RNA-induced silencing complex (RISC) (Figure 6) [157,158].

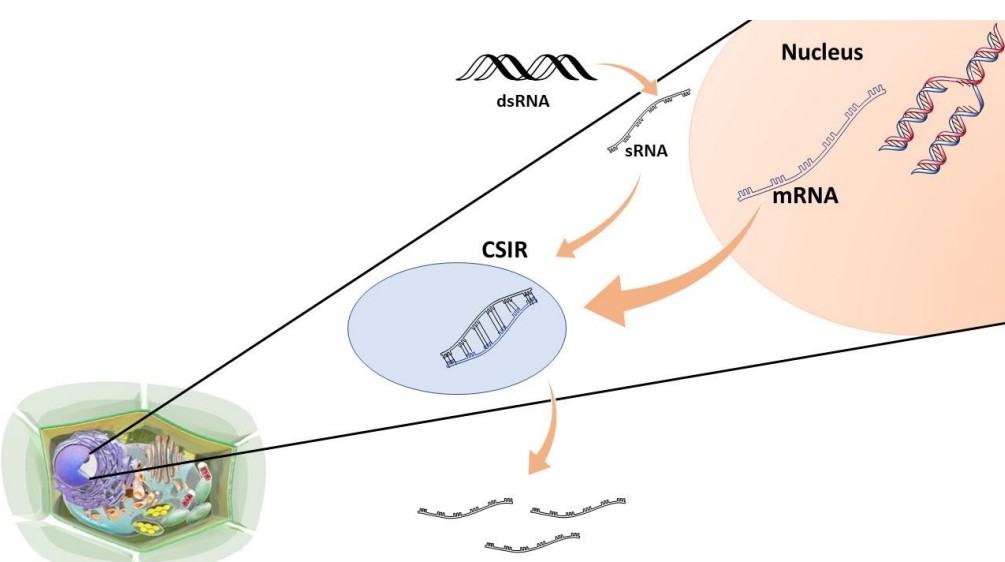

**Figure 6.** Graphical summary of cellular RNAi silencing in the mRNA target. The cell generates RNAi (sRNA) from the degradation of dsRNA. A guide chain of RNAi is incorporated into the RISC that subsequently binds to an mRNA complementary sequence, leading to degradation [159].

In recent years, genetically improved cultivars with high levels of disease resistance through the use of RNAi have been one of the alternative methods to control diseases transmitted by phytopathogenic fungi. For example, Thakur et al. (2020) reported AcCHS7 gene silencing associated with *Alternaria* leaf spot disease in *Cyamopsis tetragonoloba* [160]. Silencing the posttranscriptional gene AcCHS7 associated with mycelial growth and conidiation of *Alternaria cyamopsidis* is an efficient approach to controlling *Alternaria*. In the same way, Zhai et al. (2018) used RNAi to silence the tomato *PR5* gene for resistance to *A. alternata* [161].

Zhang et al. (2021) identified two new proteins (MdPR10-2 and MdPR10-1) associated with the pathogenesis of *A. alternata f. sp. mali*. The two proteins interact with the

resistance protein MdRNL2, consisting of a nucleotide-binding domain and leucine-rich repeats (CC-NB-LRRs) interacting with the protein MdRNL6. The two proteins form an MdRNL6–MdRNL2 protein complex which grants resistance to *A. alternata f. sp. mali* attack in *Malus* x *domestica* [162]. The production of MdPR10-2 and MdPR10-1 has been reported to significantly enhance the induced resistance in *Malus domestica* against leaf spot disease of four strains of *A. alternata f. sp. mali*: BXSB5, ALT1, BXSB7, and GBYB2. It has been suggested that this resistance occurs through the resistance induced in *Malus domestica* against *A. alternata* infection through the suppression of aldose-6-phosphate reductase (*A6PR*), generated by the regulation of 56 nucleotide-binding genes [163]. The overexpression of the resistance protein encoded by the *MdNLR16* gene in *Malus domestica* significantly improves the control of *A. alternata* disease mediated by the overexpression of the HRIP1 gene. Transcriptional regulation of *MdNLR16* occurs through binding *MdWRKY79* to the *MdNLR16* promoter in response to sorbitol. Thus, sorbitol regulates resistance to *A. alternata* infection by expressing the *MdNLR16* gene. Given the urgent need to increase synergy in sustainable management strategies against crop losses caused by phytopathogenic fungi, RNAi mechanisms play a fundamental role in increasing host resistance [164]. Phytopathogenic gene control strategies through RNAi mechanisms are interesting approaches since these are conserved mechanisms in all kingdoms, considered eco-friendly fungi control alternatives.

### 3.3.3. Control of *Alternaria* Phytopathogenicity through Regulation of Transcriptional Factors

The study of transcription factors in phytopathogenic fungi is key to regulating their pathogenicity. Transcription factors are proteins that bind to specific DNA regulatory sequences, which regulate the flow of information from DNA to messenger RNA (mRNA) [165]. Transcription factors as regulatory elements in phytopathogenic fungi play an essential role in their pathogenicity. Their ubiquity has been reported in many phytopathogenic species through fungal genome annotation [166]. Families of transcription factors in species of phytopathogenic fungi are classified according to their DNA-binding domains, which are involved in multiple stages related to infection [167].

Transcription factors consist of specific regulatory proteins that act as intermediaries in the expression of many genes associated with critical biosynthetic processes [168]. Transcription factors in fungi use cofactors such as $Zn^{2+}$, as in the case of the $Zn^{2+}Cys6$ transcription family, which is specific in families of phytopathogenic fungi, mainly in ascomycetes [169]. In this context, Chen et al. (2021) reported six transcription factors involved in the pathogenicity of the *A. alternata* mandarin pathotype (AaAreA, AaAreB, AaLreB, AaLreA, AaNsdD, and AaSreA). The study revealed a significant decrease in pathogenicity after suppressing the transcription factors AaAreA, AaAreB, AaLreB, AaLreA, AaNsdD, and AaSreA, which significantly reduced virulence. Comparative transcriptome analysis between mutants with deletions in these transcription factors revealed alterations in the expression of genes linked to oxidoreductase activity, secondary metabolism, and amino acid metabolism [170]. In addition, Gai et al. (2022) identified genes involved in transcription factors and critical metabolic pathways by analyzing the *A. alternata* transcriptome, enabling the analysis of the regulatory and biological roles of genes involved in the transcriptional factor *bZIP* by mutagenesis [171].

Additionally, Chung et al. (2020) reported the critical aspects of the transcriptional regulator *SreA* in *Alternaria alternata* strains [12]. *SreA* is linked to fungus growth and ACR toxin production, which generate necrotic spots in the vegetative tissue. Mutant strains lacking the transcriptional regulator *SreA* showed defects or severe alterations in the development and increased susceptibility in the fungal cell wall. Recently, Yang et al. (2022) identified and characterized the putative calcineurin-responsive transcription factor AaCrz1 in *A. alternata* and suggested that it might regulate the early stage of *A. alternata* colonization [172]. Fungal pathogenicity is known to evolve as a function of transcription factors, which act as fundamental regulators of gene expression and are associated with the coordinated regulation of molecular virulence [173]. The evolution of transcription factors in

fungal pathogenicity is a practical phylogenetic resource to enhance virulence investigations on key pathogen lineages.

## 4. Conclusions

In this review, we presented recent literature about eco-friendly methods for controlling the pathogenicity of *Alternaria* spp. We reviewed the available literature on essential oil and biopolymers methods, antagonist microorganisms, with particular emphasis on molecular-based methods such as CRISPR/Cas9, RNAi, and regulation of transcriptional factors available for *Alternaria* species. Despite the scarce literature, there is a growing interest in applying greener alternatives to control the diseases caused by *Alternaria* species, such as coatings of essential oils and microbial antagonists, which are safer and do not present environmental or safety concerns. In recent years, more *Alternaria* species have been identified based on molecular studies and the genome assembly according to the mycotoxins in infected hosts. The knowledge and understanding of the infection processes with the mycotoxins produced are clues for developing gene-editing strategies such as RNAi and CRISPR/Cas. Additionally, the information related to transcription factors associated with the mycotoxin genes is the clue for the editing strategy, as some studies have revealed. However, despite all the literature reviewed on gene-editing techniques for *Alternaria* species, the information is still scarce. Therefore, future research collecting genetic information on phytopathogenic species must expand the strategy. The development of new CRISPR/Cas9 systems and RNAi allows the expansion of knowledge about the biosynthetic potential of genes of interest in phytopathogenic species. In this way, it will be possible to improve the synergy of biological control alternatives through their phytopathogenic genes, substituting chemical methods to control phytopathogenic species, generating a sustainable alternative to preserve health and reduce the global incidence.

**Author Contributions:** Conceptualization, C.D.G.-T. and C.C.-L.; methodology, C.D.G.-T., D.C.C.-H. and Y.P.-R.; validation C.D.G.-T. and C.C.-L.; investigation, D.C.C.-H. and C.C.-L.; resources, E.F.-L. and C.D.G.-T.; writing—original draft preparation, C.D.G.-T. and D.C.C.-H.; writing—review and editing, D.C.C.-H., E.F.-L., C.C.-L., Y.P.-R. and C.D.G.-T.; visualization, C.D.G.-T. and C.C.-L.; supervision, C.D.G.-T., E.F.-L. and C.C.-L.; funding acquisition, C.D.G.-T. and E.F.-L. All authors have read and agreed to the published version of the manuscript.

**Funding:** This research has been funded by Dirección General de Investigaciones of Universidad Santiago de Cali under call No. 01-2022.

**Institutional Review Board Statement:** Not applicable.

**Data Availability Statement:** Not applicable.

**Conflicts of Interest:** The authors declare no conflict of interest.

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
