# Peer review of "Eco-Friendly Biocontrol Strategies of Alternaria Phytopathogen Fungus: A Focus on Gene-Editing Techniques"

_agriculture, doi:10.3390/agriculture12101722_

Round 1

Reviewer 1 Report

The manuscript entitled "Eco-friendly biocontrol strategies of Alternaria phytopathogen 2 fungus: a focus on gene-editing techniques" has been reviewed.

1.      The family name of plant or fungi is better not in italic, however the genus names must be.

2.      L116 should be revised and rewritten.

3.      The gene regions used for phylogenetic analysis of Alternaria should be correctly written with uniform abbreviations.

4.      In Table 1, the Technique for identification is not necessary to display. By the way, the descriptions are not well prepared.

5.      L151 the propagation by insects should be reconsidered and better to use some references.

6.      For Figure 2, the conidia and vegetative hyphae pictures are not Alternaria, which should be replaced. By the way, the Figure 2 is no meaning and can be removed.

7.      L 308-313 is the same as previous sentences L282-287.

8.      The authors give a comprehensive work after reviewing a number of references. However, the orders of each content should be reconsidered (2. Alternaria spp.; 2. Alternaria Infection; 3. Phytopathogenic genes in Alternaria; 4. Comtrol methods), and the titles also such as ‘Alternaria and its control strategies’.

Author Response

We profoundly appreciate all the reviewer's suggestions. They made it possible to improve the manuscript. All the suggestions were discussed point by point and addressed according to the proposal. In the manuscript, all the corrections are highlighted in blue. 

Reviewer 1

  1. The family name of plant or fungi is better not italicized; however, the genus names must be.

R// We appreciate the reviewer's suggestions. Adjustments were made throughout the document.

  1. L116 should be revised and rewritten.

R// We appreciate the reviewer's suggestions. The sentence was removed.

  1. The gene regions used for phylogenetic analysis of Alternaria should be correctly written with uniform abbreviations.

R// We appreciate the reviewer's suggestion on writing our work. Adjustments were made throughout the document.

  1. In Table 1, the Technique for identification is not necessary to display. By the way, the descriptions are not well prepared.

R// We appreciate the reviewer's correction. The "identification techniques" column was removed from table 1.

  1. L151 the propagation by insects should be reconsidered and better to use some references.

R// We appreciate the suggestion of the reviewer. We rewrite it as: "The sporulation of these fungi facilitates propagation by the breeze and through vectors such as insects, which contract the spores of phytopathogenic fungi and spread them to healthy plants [38]." Spores spread through vectors such as insects were considered in lines 149-155.

  1. For Figure 2, the conidia and vegetative hyphae pictures are not Alternaria, which should be replaced. By the way, Figure 2 is no meaning and can be removed.

R// We appreciate the reviewer's suggestion. Figure 2 was removed from the document.

  1. L 308-313 is the same as previous sentences L282-287.

R// We appreciate the reviewer's observation. The sentences of the indicated lines were revised and corrected.

  1. The authors give a comprehensive work after reviewing a number of references. However, the order of each content should be reconsidered (2. Alternaria spp.; 2. Alternaria Infection; 3. Phytopathogenic genes in Alternaria; 4. Control methods), and the titles also such as 'Alternaria and its control strategies'.

R// We appreciate the reviewer's observation. The order of the sections was revised and corrected with four units and subsections:

  1. Introduction
  2. Alternaria

2.1 Alternaria infection

  1. Alternaria control strategies

3.1 Chemical control strategies

3.2 Green control strategies

3.2.1 Essential oils used to control Alternaria fungi

3.2.2 Antagonistic microorganisms to control Alternaria

3.3. Biocontrol alternatives of Alternaria through gene editing

3.3.1 Control of Alternaria pathogenicity based on CRISPR/Cas9

3.3.2 RNAi in the control of Alternaria

3.3.3 Phytopathogenic genes in Alternaria spp.

3.3.4 Control of phytopathogenic fungi based on Transcription factors control

  1. Conclusions

Reviewer 2 Report

A review report on manuscript titled:

Eco-friendly biocontrol strategies of Alternaria phytopathogen  fungus: a focus on gene-editing techniques

General overview:

The review discussed one of the important topics about using of Eco-friendly biocontrol strategies for control and inhibition of phytopathogen fungi that pollute the agriculture and food production. A promising data were provided that Alternaria species, such as coatings of essential oils and microbial antagonists, are safer and do not present environmental or safety concerns. The knowledge and understanding of the infection processes with the mycotoxins produced are clues for developing gene-editing strategies such as RNAi and CRISPR/Cas. Also, the information related to transcription factors associated with the mycotoxin genes is the clue for the editing strategy, as some  studies have revealed. Since the frame work of the review is promising, it must be improved in its overall quality many typos errors were provided and sentences without verbs. The rigors revision is important.

Abstract:

Line 19,20 : Revise the grammar, correction again.

Line 23,24: Revise and correct; the sentence is long.

 Line 28, 32: The sentence should start by the present review for example instead of we also….

Introduction:

Line 43; correct and revise the grammar

Combine the fifth line 77 and forth paragraph together in one paragraph.

Line 91: Revise the spelling, correction again

There are some writing mistake in the format should be correct.

It is the authours’ responsibility to maintain the consistence of reference format.

Please carefully check the significant figures presented in the table.

Recommendation

As this review is promising for biological(green) control against pathogenic fungi, it should be accepted after minor revision and correction of the mentioned mistakes given in this report.

Sincerely,

Yours,

The reviewer 

Author Response

The authors deeply appreciated all the suggestions from the reviewer. The comments were addressed point by point in the attached letter and highlighted in the manuscript in blue.

 Reviewer 2

A review report on the manuscript titled: Eco-friendly biocontrol strategies of Alternaria phytopathogen fungus: a focus on gene-editing techniques

General overview:

The review discussed one of the important topics about using of Eco-friendly biocontrol strategies for control and inhibition of phytopathogen fungi that pollute the agriculture and food production. A promising data were provided that Alternaria species, such as coatings of essential oils and microbial antagonists, are safer and do not present environmental or safety concerns. The knowledge and understanding of the infection processes with the mycotoxins produced are clues for developing gene-editing strategies such as RNAi and CRISPR/Cas. Also, the information related to transcription factors associated with the mycotoxin genes is the clue for the editing strategy, as some studies have revealed. Since the framework of the review is promising, it must be improved in its overall quality many typos errors were provided and sentences without verbs. The rigors revision is important.

Abstract:

Line 19,20: Revise the grammar, correction again.

R// We appreciate the reviewer's suggestion. The writing was reviewed and corrected in lines 19-22 as: " In this context, the control of diseases such as early blight caused by Alternaria solani in potatoes and Alternaria linariae in tomatoes has mainly consisted of the application of fungicides, with negative impacts on the environment and human health. Recently, the application of ʹomicʹ and gene editing through the CRISPR/Cas9 system and RNAi technologies demonstrate their effectiveness as emerging greener alternatives for controlling phytopathogenic fungi. Additionally, coatings based on essential oils and microbial antagonists suggest alternative strategies for controlling phytopathogenic fungi that are respectful of the environment."

Line 23,24: Revise and correct; the sentence is long.

R// We appreciate the reviewer's suggestion. The text was shortened and separated into two sentences: "Recently, the application of ʹomicʹ and gene editing through the CRISPR/Cas9 system and RNAi technologies demonstrate their effectiveness as emerging greener alternatives for the control of phytopathogenic fungi. Additionally, coatings based on essential oils and microbial antagonists suggest alternative strategies for controlling phytopathogenic fungi that are respectful of the environment."

 Line 28, 32: The sentence should start by the present review for example instead of we also….

R// We appreciate the reviewer's suggestion. The text was corrected as suggested changing the personal pronouns.

Introduction:

Line 43; correct and revise the grammar

R// We appreciate the reviewer's suggestion. The text was corrected: "Some estimates of losses caused by phytopathogenic fungi in crops are 21.5% for wheat, 30% for rice, 22.5% for corn, 17.2% for potatoes, and 21.4% for soybeans, especially in the postharvest production stage."

Combine the fifth line 77 and the fourth paragraph together in one paragraph.

R// We appreciate the reviewer's suggestion. The line was combined with the previous paragraph.

Line 91: Revise the spelling, correction again

R// We appreciate the reviewer's suggestion. The text was corrected: "Other eco-friendly alternatives for phytopathogenic fungi management include essential oils and microbial antagonists."

There are some writing mistakes in the format that should be corrected.

R// We appreciate the reviewer's suggestion. The manuscript was carefully checked, and all the typos and grammar mistakes were corrected.

It is the author's responsibility to maintain the consistency of the reference format.

R// We appreciate the reviewer's comment. The manuscript was carefully checked, and all the typos and grammar mistakes were corrected.

Please carefully check the significant figures presented in the table.

R// We appreciate the reviewer's comment. All the tables were carefully checked and corrected for significant figures.

Recommendation

As this review is promising for biological(green) control against pathogenic fungi, it should be accepted after minor revision and correction of the mentioned mistakes in this report.

R// We sincerely thank the positive comment for the reviewer. We tried to address all the suggestions to improve the manuscript. It is crucial for us, the reviewers put, to improve the work.

Reviewer 3 Report

In this review the authors have shown different strategies particularly gene editing used to confront the phytopathogen Alternaria sp. The review is quite comprehensive and well written. The literature presented here will be useful to the researchers working on these perspective. Since this review paper is based only on single phytopathogen I would suggest authors to cover all the examples till date pertaining to aspects mentioned in the paper. Overall manuscript is well presented and I would recommend it to be considered for publication in reputed journal “Agriculture”. However, there are few suggestions that I feel authors should address (mentioned below)

1. It would be better if different types of TCG genes identified in Alternaria sp. so far should be written. in line no. 56

2. In the introduction part line no. 57 to 70 authors have given the examples about the strategies used for regulating pathogens. These examples should be written later while discussing CRISPR/CAS technique (section 5.1). In the introduction general overview should be provided.

3. In line no. 71. Authors should define briefly about Biosynthetic genes. Why these should be targeted?

4. In section 4.1 (line no. 259-262) Are the compounds which have been exemplified in the section have been used to control Alternaria? If this is the case then give reference in support of these.

6. In section 4.3 few lines should be written regarding the mode of action of antagonistic organism on phytopathogens. (For eg. Competition for nutrients, space, secondary metabolites)

7. The title of section 5 “Biocontrol alternatives through molecular studies” is a bit confusing. It should be modified. This section covers the molecular based methods for controlling phytopathogen. So should be changed accordingly.

8. The section 6. Phytopathogenic genes in Alternaria spp should come before control methods.

Section 7 should come in continuation with control methods

9. Initial lines of conclusion needs to be paraphrased

Author Response

We are profoundly thankful for all the constructive comments and suggestions to the manuscript that helped us to improve the quality and understanding. All the requests were carefully addressed, as you can verify in the answer letter and the modified version of our manuscript. 

Reviewer 3

Comments and Suggestions for Authors

In this review the authors have shown different strategies particularly gene editing used to confront the phytopathogen Alternaria sp. The review is quite comprehensive and well written. The literature presented here will be useful to the researchers working on these perspective. Since this review paper is based only on single phytopathogen I would suggest authors to cover all the examples till date pertaining to aspects mentioned in the paper. Overall manuscript is well presented and I would recommend it to be considered for publication in reputed journal "Agriculture". However, there are few suggestions that I feel authors should address (mentioned below)

  1. It would be better if different types of TCG genes identified in Alternaria sp. so far should be written. in line no. 56.

R// We appreciate the reviewer's comment. We modified the text between lines 60-64 as: “In recent years, the study of TCG such as Polyketide Synthase A (PksA), associated with protein biosynthesis, as well as the AMT, AFT, AKT, and ALT gene groups, associated with pathogenicity and host specificity in A. alternata [9], has allowed the development of strategies based on gene silencing or activation, taking advantage of the CRISPR/Cas9 system [10].”

  1. In the introduction part, line no. 57 to 70 authors have given examples of

the strategies used for regulating pathogens. These examples should be written later while discussing CRISPR/CAS technique (section 5.1). In the introduction general overview should be provided.

R// We appreciate the reviewer's comment. We moved the paragraph to section 3.3 "Molecular-based methods for controlling Alternata," between lines 436-442 and 613-617. We did not introduce it precisely in the CRISPR/Cas section since the gene disruption was performed by knockout gene-targeted techniques, and the other information was related to the regulation of pathogenicity of A. alternata by regulation of transcriptional factors.

  1. In line no. 71. Authors should define briefly about Biosynthetic genes. Why these should be targeted?

R// We appreciate the reviewer's suggestion. Biosynthetic phytopathogenic genes were briefly defined (since they are the target) in lines 72-80. The paragraph was modified as follows: “Biosynthetic genes are interesting in developing alternative pathogenicity control methods for Alternaria (and other phytopathogenic fungi), especially phytopathogenic genes, which are producers of various secondary metabolites (small molecules with biological activity in the organism) that are closely related to the various interactions with other organisms and are located (clustered) adjacently in the genomes of various plant pathogenic species [13]. These groups of genes associated with the production of secondary metabolites have been marked as targets for gene editing methodologies, given that these genes are related to the fungal chemodiversity of various phytopathogenic species, as well as the functional variety and horizontal transfer.”

  1. In section 4.1 (line no. 259-262) Are the compounds which have been exemplified in the section have been used to control Alternaria? If this is the case then give reference in support of these.

R// We appreciate the suggestion from the reviewer. The essential oils and antagonistic microorganisms reported in tables 2 and 3 were separated into different rows, according to the reporting studies, with their respective references.

  1. In section 4.3 few lines should be written regarding the mode of action of antagonistic organism on phytopathogens. (For eg. Competition for nutrients, space, secondary metabolites)

R// We appreciate the reviewer's suggestion. A brief description was added on the competition for nutrients of the antagonistic microorganisms in lines 398-405: “which are based on mycoparasitism or direct parasitism through bacteria. Bacterial-based parasitism has been studied in previous publications, which report that these associations between antagonistic microorganisms and phytopathogenic fungi are based on the adherence of antagonistic microorganisms to both the conidia and vegetative hyphae of the fungus, which aggressively compete for nutrients [116]. These bacteria have been identified as species belonging to the Bacillus genus, such as B. cereus, B. mycide, and B. subtilis, which have shown high efficiency in controlling Alternaria helianthi.”

  1. The title of section 5 "Biocontrol alternatives through molecular studies" is a bit confusing. It should be modified. This section covers molecular-based methods for controlling phytopathogen. So, it should be changed accordingly.

R// We appreciate the reviewer's suggestion. The title of section 3.3 on line 431 was modified to: "Molecular-based methods for controlling Alternaria."

  1. The section 6. Phytopathogenic genes in Alternaria spp should come before control methods.

R// We appreciate the reviewer's suggestion. The order was changed to the reviewer's one suggestion which coincided with this suggestion.

Section 7 should come in continuation with control methods

R// We appreciate the reviewer's suggestion. The order was changed to the reviewer's one suggestion which coincided with this suggestion.

  1. Initial lines of the conclusion needs to be paraphrased

R// We appreciate the reviewer's suggestion. We paraphrased the title and some lines as an introduction for the conclusion section (lines 628-632).

Round 2

Reviewer 1 Report

Dear editor:

  The manuscript entitled "Eco-friendly biocontrol strategies of Alternaria phytopathogen 2 fungus: a focus on gene-editing techniques " has been reviewed again. The manuscript has properly revised. There are some places need to be considered for revisions (Minor).

1.     L 104 ‘Alternaria spp.’ Alternaria should be in italic.

2.     L 111 Altenaria was firstly described by Nees in 1816, not 1817.

3.     L 453 f. sp. should be in italic.

4.     L 239 3.2.1 ‘Essential oils’ should be replaced by ‘Essential oils and biopolymers’.

5.  In my opinion, the order of this manuscript could be as following:

2. Alternaria spp.

2.1 Alternaria infection

2.2 Phytopathogenic genes in Alternaria

2.3 Transcription factors in Alternaria

3. Alternaria control strategies

3.1 Chemical control strategies

3.2 Green control strategies

3.2.1 Essential oils used to control Alternaria fungi

3.2.2 Antagonistic microorganisms to control Alternaria

3.3. Biocontrol alternatives of Alternaria through gene editing

3.3.1 Control of Alternaria pathogenicity based on CRISPR/Cas9

3.3.2 RNAi in the control of Alternaria

6. By the way, the whole manuscript should be carefully check after revising, and the logical connection between each paragraph should be carefully re-checked.

Have a great day!

Author Response

We are profoundly thankful for all the constructive comments and suggestions to the manuscript that helped us to improve the quality and understanding. All the requests were carefully addressed, as you can verify in the answer letter and the modified version of our manuscript. 

Reviewer 1

Comments and Suggestions for Authors

The manuscript entitled "Eco-friendly biocontrol strategies of Alternaria phytopathogen 2 fungus: a focus on gene-editing techniques " has been reviewed again. The manuscript has properly revised. There are some places need to be considered for revisions (Minor).

  1. L 104 ‘Alternaria spp.’ Alternaria should be in italic.

R// We appreciate the reviewer's suggestions. The name was changed to italics on line 113 (section 2) as "Alternaria spp."

  1. L 111 Alternaria was firstly described by Nees in 1816, not 1817.

R// We appreciate the reviewer's suggestion. "The description of the Alternaria genus dates to 1816 with the first type of Alternaria tenuis identification." The date was corrected in line 120.

  1. L 453 f. sp. should be in italic.

R// We appreciate the reviewer's suggestion. The name was corrected to italics in line 565.

  1. L 239 3.2.1 'Essential oils' should be replaced by 'Essential oils and biopolymers.'

R// We appreciate the suggestion from the reviewer. The title of section 3.2.1 was corrected to: "Essential oils and biopolymers used to control Alternaria fungi."

  1. In my opinion, the order of this manuscript could be as follows:

R// We sincerely appreciate the reviewer's suggestion. It helped to clarify the logical order of our work. However, we did not move the section related to the regulation of transcriptional factors since it is related to the molecular-based control. The specific order of the sections was:

  1. Alternaria spp.

2.1 Alternaria infection

2.2 Phytopathogenic genes in Alternaria

  1. Alternaria control strategies

3.1 Chemical control strategies

3.2 Green control strategies

3.2.1 Essential oils and biopolymers used to control Alternaria fungi

3.2.2 Antagonistic microorganisms to control Alternaria

3.3 Molecular-based methods for controlling Alternaria

3.3.1 Control of Alternaria pathogenicity based on CRISPR/Cas9

3.3.2 RNAi in the control of Alternaria

3.3.3 Control of Alternaria phytopatogenecity through regulation of transcriptional factors

  1. By the way, the whole manuscript should be carefully checked after revising, and the logical connection between each paragraph should be carefully re-checked.

R// We sincerely appreciate the reviewer's suggestion. We organized ideas and paragraphs properly. The fluency and connection between each paragraph were checked and corrected.
